

# Association of time-averaged serum uric acid level with clinicopathological information and long-term outcomes in patients with IgA nephropathy

Mengjie Weng[1,2,3,*], Binbin Fu[1,2,3,*], Yongjie Zhuo[1,2,3], Jiaqun Lin[1,2,3], Zhenhuan Zou[1,2,3], Yi Chen[1,2,3], Jiong Cui[1,2,3], Guifen Li[1,2,3], Caiming Chen[1,2,3], Yanfang Xu[1,2,3], Dewen Jiang[1,2,3] and Jianxin Wan[1,2,3]

[1] Department of Nephrology, Blood Purification Research Center, The First Affiliated Hospital of Fujian Medical University, Fuzhou, Fujian, China

[2] Fujian Clinical Research Center for Metabolic Chronic Kidney Disease, The First Affiliated Hospital of Fujian Medical University, Fuzhou, Fujian, China

[3] Department of Nephrology, National Regional Medical Center, Binhai Campus of the First Affiliated Hospital, Fujian Medical University, Fuzhou, Fujian, China

[*] These authors contributed equally to this work.

Corresponding author
Jianxin Wan, wanjx@fjmu.edu.cn

## ABSTRACT

**Objective**. Whether serum uric acid (SUA) at baseline could been identiûed as a risk factor for progression in IgA nephropathy (IgAN) patients remains unclear, therefore, long- term SUA control levels must be monitored. We aimed to investigate the relevant factors affecting time-averaged SUA (TA-SUA) and to assess the prognostic value of TA-SUA in IgAN.

**Methods**. This retrospective study included 152 patients with IgAN. The relationships between TA-SUA and clinicopathological features and renal outcomes (defined as the doubling of the baseline serum creatinine level or end-stage renal disease) were analyzed in groups divided by quartiles of TA-SUA levels, the presence of hyperuricemia, and sex.

**Results**. Patients with high TA-SUA levels had higher levels of baseline SUA, blood urea nitrogen (BUN), triglycerides, serum C3 and serum C4 and were more likely to be male and have hypertension, proteinuria, poor renal function, and pathological injuries including high grades of tubular atrophy/interstitial fibrosis (T1–T2). These patients had a poorer prognosis compared with patients with low TA-SUA levels. The TA-SUA level was positively correlated with baseline age and BUN, triglycerides, serum C3, and serum C4 levels, and negatively correlated with baseline eGFR. Survival curve analysis indicated that persistent hyperuricemia was associated with significantly poorer renal outcomes than normo-uricemia in both men and women. The TA-SUA level also was an independent predictor of renal outcome in patients with IgAN, with optimal cutoû values of 451.38 $\mu$mol/L (area under the curve (AUC) = 0.934) for men and 492.83 $\mu$mol/L (AUC = 0.768) for women.

**Conclusions**. The TA-SUA level is associated with triglyceride level, complement component levels, renal function, and pathological severity of IgAN, and it may be a prognostic indicator in male and female patients with IgAN.

## INTRODUCTION

IgA nephropathy (IgAN) is a common primary glomerulonephritis disease. and affects an estimated 200,000–350,000 people per year globally (*Schena & Nistor, 2018*). Its diagnosis is associated with a reduction in life expectancy by 6–10 years (*Jarrick et al., 2019*). It has previously been observed that renin-angiotensin-aldosterone system (RAAS) inhibitors were effective for long-term renal survival of advanced IgAN, although proteinuria and blood pressure did not decrease (*Moriyama et al., 2011*), but approximately 30% of patients with IgAN who are older than 30 years will develop terminal chronic renal failure within 20 years after diagnosis (*D'Amico, 2004*). Therefore, primary prevention strategies, including early detection and strategies to control risk factors, are needed to protect against the poor renal outcomes of IgAN and its related complications. Recent research demonstrated that the new Oxford classification of IgAN is valuable for predicting the prognosis of IgAN and created an updated MEST-C scoring system that includes mesangial hypercellularity (M), endocapillary hypercellularity (E), segmental glomerulosclerosis (S), tubular atrophy/interstitial fibrosis (T) and crescents (C) (*Barbour et al., 2016*; *Trimarchi et al., 2017*). Previous research has identified several risk factors for the progression of IgAN to chronic renal failure or end-stage renal disease (ESRD), including an elevated serum creatinine (SCr) level, hypertension, proteinuria, dyslipidemia and hyperuricemia (*Goto et al., 2009*; *Syrjänen, Mustonen & Pasternack, 2000*).

However, the contribution of the serum uric acid (SUA) level in IgAN progression remains controversial. Some previous studies found that hyperuricemia may be an independent risk factor for ESRD development in patients with IgAN (*Moriyama et al., 2014*; *Lu et al., 2020*; *Geng et al., 2022*), whereas other studies found that the relationship between hyperuricemia and IgAN progression was not very significant in patients with older age, lower estimated glomerular filtration rate (eGFR), or interstitial lesion (*Zhu et al., 2018*). Furthermore, additional studies have suggested that the SUA level is a predictor of IgAN only in female patients and not in male patients (*Nagasawa et al., 2016*; *Oh et al., 2020*) and that hyperuricemia is only a risk factor for the progression of IgAN in patients with stage G3a chronic kidney disease (CKD) (*Moriyama et al., 2015*). The existing research on the relationship between SUA and the progress of IgAN is not very consistent. One major reason for these conflicting results may be that a single measurement of SUA, as employed in these studies, cannot reflect the longitudinal variation and cumulative burden associated with elevated SUA levels. Additionally, as the condition progresses, the SUA levels alter continually. These limitations and inconsistencies have restricted efforts to analyze associations between the SUA level and long-term outcomes.

Therefore, the concept of time-averaged serum uric acid (TA-SUA), as a measurement representing the intensity of SUA during follow-up, was applied in the present study. TA-SUA has been used previously in analyses of blood pressure, hematuria and proteinuria in several cohort studies (*Sevillano et al., 2017*; *Yu et al., 2020*). In this retrospective

study, we evaluated the clinicopathological characteristics of four groups of patients with IgAN defined according to quartiles of TA-SUA during follow-up and investigated the association of persistent hyperuricemia with the clinicopathological severity and long-term renal outcomes of IgAN in male and female patients. Receiver-operating characteristic (ROC) curve analysis was performed to determine the predictive value of TA-SUA for the progression of IgAN. We hypothesized that persistent hyperuricemia was significantly related to the clinicopathological severity and long-term renal prognosis of male and female patients with IgAN, and the TA-SUA index reflecting persistent hyperuricemia could be used as a predictor of the progress of IgAN.

## METHODS

### Patients and samples

This cohort study included IgAN patients who underwent kidney biopsy between June 2012 and December 2018 in the Department of Nephrology of the First Affiliated Hospital of Fujian Medical University. The study adhered to all relevant tenets of the Declaration of Helsinki and was approved by the Ethics Committee of the First Affiliated Hospital of Fujian Medical University (MTCA, ECFAH of FMU [2015] 084-2). Informed written consent was obtained from all patients. The exclusion criteria included: age <18 years; follow-up <12 months; and a secondary cause of IgAN, such as liver or inflammatory bowel diseases, inflammatory chronic disease, other autoimmune disorder, active cancer, and Henoch-Schönlein purpura. All patients were followed up until June 2021. Finally, a total of 152 patients were included in this study (Fig. 1).

### Patient follow-up and data collection

All patients were followed up through regular visits at intervals of 6 months. Serum uric acid, urine sediment, serum albumin, and serum creatinine (SCr) were tested at every visit and only the 24 h urinary protein was tested at first 6 months. Information regarding medication use was collected during follow-up, including the use of a RAAS inhibitor, a glucocorticoid, and an immunosuppressant (including tacrolimus, cyclophosphamide, and cyclosporine).

The clinical data of enrolled patients were collected at the time of renal biopsy and included each patient's sex, age, serum albumin level, SCr level, estimated glomerular filtration rate (eGFR), blood urea nitrogen (BUN) level, serum uric acid level, hemoglobin level, total cholesterol (TC) level, triglyceride (TG) level, serum C3 level, serum C4 level, 24-h urinary protein level, urinary red blood cell count, and any history of macroscopic hematuria, diabetes and hypertension. Pathologic lesions were evaluated according to the new Oxford classification (MEST-C) (Trimarchi et al., 2017), including whether the mesangial score was ≤0.5 (M0) or >0.5 (M1); segmental glomerulosclerosis was absent (S0) or present (S1); endocapillary hypercellularity was absent (E0) or present (E1); tubular atrophy atrophy/interstitial fibrosis was ≤25% (T0), 26–50% (T1) or >50% (T2); and

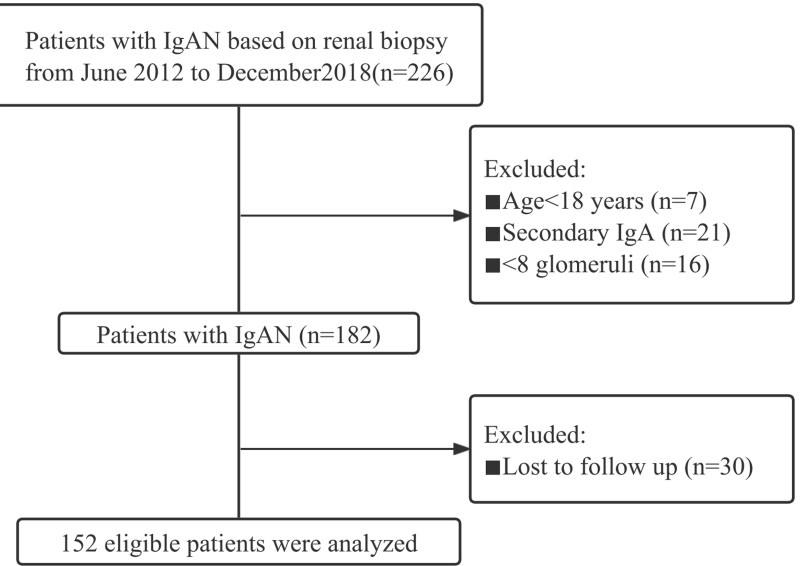

**Figure 1    The study flow chart of the enrolment of IgAN patients.**

cellular/fibrocellular crescents were absent (C0), present in at least 1 glomerulus (C1), or present in >25% of glomeruli (C2).

The composite endpoint was the doubling of the baseline SCr level (D-SCr) or ESRD, indicated by an eGFR <15 ml/min per 1.73 m$^2$ or the initiation of renal replacement therapy.

## The TA-SUA

For each patient, summary measures (time-averaged) were calculated for serum uric acid level, serum albumin level, and microscopic hematuria from the area under the curve (AUC) of serial measurements standardized by the length of the study (*Le et al., 2012*). The TA-SUA was defined as the AUC for SUA during follow-up divided by the total number of months of follow-up. Similarly, the time-averaged serum albumin (TA-serum albumin) and time-averaged hematuria (TA-hematuria) values were calculated with the same method used for TA-SUA. Fig. S1 outlines the serial measurement of SUA during follow-up for one of the patients in this study. Patients were categorized into two groups according to the degree of TA-SUA: persistent hyperuricemia and normo-uricemia. The present study employed the established definition for hyperuricemia of a SUA level ≥420 μmol/L for men and ≥360 μmol/L for women (*Iseki et al., 2004*). Normo-uricemia was defined by TA-SUA values of <420 μmol/L in men and <360 μmol/L in women.

## Hypertension and macroscopic hematuria

Hypertension was defined as a systolic blood pressure ≥140 mmHg or diastolic blood pressure ≥90 mmHg measured at rest on 3 different days, or a diagnosis of hypertension that was controlled with antihypertensive medication. Macroscopic hematuria was defined by urine that was visibly tea-colored, cola-colored, pink, or even red.

## Proteinuria remission and eGFR

Proteinuria remission was defined as proteinuria with protein excretion ≤1.0 g/d and a decline of more than 25% of the base value (*Canney et al., 2021*), and all cases not meeting these criteria were considered to have persistent proteinuria. The eGFR was calculated using the CKD-EPI formula (*Levey et al., 2009*).

## Statistical analysis

All data were analyzed using SPSS software 25.0 and GraphPad Prism 8.0. Continuous variables are presented as the mean ± standard deviation or median with quartile range. Comparisons between groups were carried out using Student's *t*-test or analysis of variance (ANOVA) and Kruskal–Wallis test. Categorical variables are described as frequency with percentage and were compared using the Chi-square test. Spearman's correlation analysis was performed to evaluate the associations between the TA-SUA level and clinical parameters. Survival curves for the groups with persistent hyperuricemia and normo-uricemia were generated using the Kaplan–Meier method. Cox proportional hazard regression was applied to calculate the hazard ratio (HR) and 95% confidence interval (CI) for the risk of IgAN. Receiver-operating characteristic (ROC) curve analysis of TA-SUA was performed to determine the optimal cutoff value for predicting a combined event. Statistical significance was defined by a *P*-value <0.05.

# RESULTS

## Baseline characteristics of patients according to quartiles of TA-SUA

Table 1 presents the characteristics of study patients based on quartiles of TA-SUA levels. The enrolled patients had a mean age of 35 (28-44) years, and 50% of them were men. The mean baseline SUA and TA-SUA levels were 392.00 (314.30–454.52) μmol/L and 376.02 (311.50–444.46) μmol/L, respectively. Patients with higher TA-SUA levels were more likely to be male and to have hypertension, and these patients also had higher levels of baseline SUA, BUN and TG, more proteinuria, worse renal function and CKD stage, more complement activation including higher levels of serum C3 and serum C4, and more serious renal pathological injuries including a higher grade of tubular atrophy/interstitial fibrosis, compared with patients with TA-SUA levels in the lowest quartile. No significant differences were detected among the four groups in age, the presence of macroscopic hematuria and diabetes, the levels of serum albumin, hemoglobin, TC and serum IgG, IgA, IgM or in the four pathological features, including M1, E1, S1 and C1–C2 (all *P* > 0.05).

During follow-up, we found that patients with TA-SUA levels in the highest quartile had more chances to receive RAAS inhibitor intervention (*P* > 0.05). However, no significant differences were observed among the four groups in the levels of TA-hematuria and TA-serum albumin during follow-up (Fig. S2). In addition, the presence of proteinuria remission during first 6 months and use of a glucocorticoid in patients with IgAN were similar in the four groups (all *P* > 0.05).

## Gender-specific characteristics according to TA-SUA levels

The baseline clinicopathological characteristics of the enrolled men and women with IgAN are listed separately in Table 2. The incidence rate of IgAN complicated with persistent

**Table 1  Baseline characteristics of patients with IgAN according to quartiles of TA-SUA.**

| | Overall | TA-SUA (µmol/L) during follow-up | | | | *P*-value |
|---|---|---|---|---|---|---|
| | | Quartile 1 (<311.50) | Quartile 2 (311.50–376.02) | Quartile 3 (376.03–444.46) | Quartile 4 (>444.46) | |
| *N* (%) | 152 | 38 | 38 | 38 | 38 | |
| Clinical characteristics | | | | | | |
| Male sex (%) | 76 (50) | 7 (18.4)[****] | 13 (34.2)[****] | 30 (78.9)[a,b] | 26 (68.4)[a,b] | <0.001[*] |
| Age (years) | 35 (28–44) | 31 (26–39) | 34 (28–45) | 34 (26.75–49.25) | 38 (30–45) | 0.230 |
| Hypertension (%) | 55 (36.2) | 5 (13.2)[****] | 11 (28.9) | 17 (44.7)[**] | 22 (57.9)[a,b] | <0.001[*] |
| Diabetes (%) | 8 (5.3) | 1 (2.6) | 1 (2.6) | 3 (7.9) | 3 (7.9) | 0.891 |
| Hyperuricemia (%) | 74 (48.7) | 1 (2.6)[b,c] | 15 (39.5)[a,c] | 23 (60.5)[a,b] | 35 (92.1)[a,b,c] | <0.001[*] |
| SUA (µmol/L) | 392.00 (314.30–454.52) | 260.00 (234.50–301.65)[b,c] | 357.80 (326.45–407.93)[a,c] | 410.70 (383.33–447.58)[a,b] | 483.00 (445.50–575.80)[a,b,c] | <0.001[*] |
| Macroscopic hematuria (%) | 58 (38.2) | 18 (47.4) | 14 (36.8) | 11 (28.9) | 15 (39.5) | 0.425 |
| BUN (mmol/L) | 5.22 (4.11–6.89) | 4.46 (3.74–4.99)[****] | 5.36 (4.01–6.29) | 5.72 (4.74–6.88)[**] | 6.98 (5.38–9.24)[a,b,c] | <0.001[*] |
| Serum albumin (g/L) | 38.55 (34.20–41.65) | 38.90 (34.15–42.60) | 38.40 (33.60–40.90) | 38.70 (36.10–41.93) | 37.50 (32.05–40.40) | 0.401 |
| 24 h urinary protein (g/d) | 1.11 (0.44–2.02) | 0.58 (0.23–1.29)[***] | 0.86 (0.30–1.92)[**] | 1.10 (0.69–2.12) | 1.75 (1.04–3.97)[a,c] | <0.001[*] |
| Hemoglobin (g/L) | 128.78 ±20.08 | 128.50 ±15.58 | 126.39 ±23.36 | 133.45 ±18.45 | 126.76 ±21.99 | 0.398 |
| TG (mmol/L) | 1.32 (0.86–2.01) | 0.92 (0.63–1.46)[****] | 1.22 (0.82–2.06) | 1.49 (1.05–1.96)[**] | 1.77 (1.18–2.64)[**] | <0.001[*] |
| TC (mmol/L) | 4.89 (4.15–5.80) | 4.49 (4.18–5.43) | 4.89 (4.13–5.99) | 4.61 (4.24–5.08) | 4.60 (4.27–5.20) | 0.200 |
| Serum C3 (g/L) | 0.97 (0.82–1.13) | 0.94 (0.81–1.03) | 0.94 (0.80–1.14) | 0.96 (0.81–1.13) | 1.06 (0.91–1.17)[a,b,c] | 0.036[*] |
| Serum C4 (g/L) | 0.22 (0.18–0.29) | 0.19 (0.16–0.26) | 0.22 (0.19–0.26) | 0.22 (0.18–0.29) | 0.26 (0.21–0.31)[**] | 0.020[*] |
| Serum IgG (g/L) | 10.59 ±3.15 | 10.57 ±3.67 | 11.01 ±2.99 | 10.06 ±3.10 | 10.71 ±2.68 | 0.613 |
| Serum IgA (g/L) | 3.09 ±0.93 | 2.78 ±0.74 | 3.19 ±1.02 | 3.24 ±1.00 | 3.17 ±0.96 | 0.122 |
| Serum IgM (g/L) | 1.27 ±0.67 | 1.37 ±0.75 | 1.44 ±0.72 | 1.13 ±0.55 | 1.12 ±0.61 | 0.080 |
| eGFR (ml/min 1.73 m²) | 90.50 (60.30–115.50) | 115.50 (100.00–129.00)[b,c] | 106.00 (82.33–117.43)[**] | 82.85 (65.40–99.38)[**] | 58.60 (42.20–76.85)[a,b,c] | <0.001[*] |
| CKD stage (1/2/3/4) | 76/39/29/8 | 34/4/0/0 | 24/6/6/2[**] | 14/15/8/1[**] | 4/14/15/5[a,b,c] | <0.001[*] |

*(continued on next page)*

**Table 1** (*continued*)

| | Overall | TA-SUA (μmol/L) during follow-up | | | | P-value |
|---|---|---|---|---|---|---|
| | | Quartile 1 (<311.50) | Quartile 2 (311.50–376.02) | Quartile 3 (376.03–444.46) | Quartile 4 (>444.46) | |
| Therapy status | | | | | | |
| RAAS inhibitor (%) | 95 (62.5) | 21 (55.3) | 14 (36.8) | 32 (84.2)[a,b] | 28 (73.7)[***] | <0.001[*] |
| Glucocorticoid (%) | 55 (36.2) | 13 (34.2) | 11 (28.9) | 14 (36.8) | 17 (44.7) | 0.544 |
| Immunosuppressant (%) | 31 (20.4) | 7 (18.4) | 6 (15.8) | 8 (21.1) | 10 (26.3) | 0.701 |
| Oxford classification (MEST-C) | | | | | | |
| M1 | 110 (72.4) | 29 (76.3) | 26 (68.4) | 28 (73.7) | 27 (27.5) | 0.883 |
| E1 | 67 (44.1) | 17 (44.7) | 14 (36.8) | 17 (44.7) | 19 (50.0) | 0.715 |
| S1 | 110 (72.4) | 26 (68.4) | 26 (68.4) | 26 (68.4) | 32 (84.2) | 0.314 |
| T1–T2 | 36 (23.7) | 3 (7.9) | 8 (21.1) | 10 (26.3) | 15 (39.5)[**] | 0.013[*] |
| C1–C2 | 54 (15.5) | 16 (42.1) | 13 (34.2) | 14 (36.8) | 11 (28.9) | 0.684 |

**Notes.**

Data are expressed as numbers and percentages in non-continuous variables, as means ±standard deviation in parametric continuous variables, and as median and interquartile range in nonparametric continuous variables.

BUN, blood urea nitrogen; SUA, serum uric acid; TG, triglycerides; TC, total cholesterol; C3, complement 3; C4, complement 4; eGFR, estimated glomerular filtration rate; CKD, chronic kidney disease; RAAS, renin-angiotensin-aldosterone system; M1, mesangial hypercellularity; E1, endocapillary hypercellularity; S1, segmental glomerulosclerosis; T1–T2, tubular atrophy/interstitial fibrosis >25%; C1–C2, presence of crescent.

[*]Two-tailed $P < 0.05$.
[**]$P < 0.05$ vs Quartile 1.
[***]$P < 0.05$ vs Quartile 2.
[****]$P < 0.05$ vs Quartile 3.

hyperuricemia was 42.1% in males and 31.6% in females during follow-up. Among both men and women, patients with persistent hyperuricemia had a significantly higher rate of hypertension (65.6%), a higher level of baseline SUA, and a lower baseline eGFR than those in the normo-uricemia groups (TA-SUA ≤ 420 μmol/L; all $P < 0.05$ for both groups). In addition, only female patients with persistent hyperuricemia had higher levels of BUN, 24-hour urine protein and TG as well as a lower level of serum albumin than those in the normo-uricemia group (all $P < 0.05$), whereas these differences were not detected between the two groups of male patients.

Among the analyzed pathological features, statistically significant trends were observed for a greater prevalence of endocapillary hypercellularity (E1) and the presence of tubular atrophy/interstitial fibrosis (T1–T2) in men with persistent hyperuricemia compared with those with normo-uricemia (both $P < 0.05$) (Table 2).

## Associations between TA-SUA level and clinical parameters

The observed correlations between the TA-SUA level and clinical parameters in patients with IgAN are presented in Table 3. The TA-SUA level was positively correlated with baseline age ($r = 0.170$, $P = 0.036$). In addition, the TA-SUA level showed strong positive correlations with the levels of BUN ($r = 0.488$, $P < 0.001$) and TG ($r = 0.359$, $P < 0.001$), as well as a strong negative correlation with eGFR ($r = -0.627$, $P < 0.001$). All of these correlations remained significant in female patients (Table 3), but the correlations of

**Table 2  Comparison of clinicopathological data between persistent hyperuricemia and normo-uricemia groups of male and female IgAN patients.**

| | Male | | | | Female | | | |
|---|---|---|---|---|---|---|---|---|
| | Overall | Persistent hyperuricemia (TA-SUA > 420) | Normo-uricemia (TA-SUA ≤ 420) | *P*-value | Overall | Persistent hyperuricemia (TA-SUA > 360) | Normo-uricemia (TA-SUA ≤ 360) | *P*-value |
| N (%) | 76 | 32 (42.1) | 44 (57.9) | | 76 | 24 (31.6) | 52 (68.2) | |
| Clinical characteristics | | | | | | | | |
| Age (years) | 37.50 (28.25–46.00) | 41.00 (30.00–46.00) | 30.50 (27.00–46.75) | 0.391 | 34.00 (28.00–41.75) | 37.00 (29.25–49.75) | 33.00 (27.00–39.00) | 0.055 |
| Hypertension (%) | 34 (44.7) | 21 (65.6) | 13 (29.5) | 0.002[*] | 21 (27.6) | 12 (50.0) | 9 (17.3) | 0.003[*] |
| SUA (μmol/L) | 427.00 (373.20–494.53) | 474.00 (437.08–520.45) | 393.90 (334.38–442.50) | <0.001[*] | 333.75 (269.20–414.35) | 477.95 (430.95–629.40) | 340.60 (300.00–400.20) | <0.001[*] |
| BUN (mmol/L) | 5.91 (4.75–7.24) | 6.75 (4.48–8.23) | 5.68 (4.92–6.26) | 0.131 | 4.80 (3.87–6.43) | 6.66 (5.07–8.45) | 4.30 (3.61–5.00) | <0.001[*] |
| Serum albumin (g/L) | 38.65 (34.28–42.38) | 38.60 (34.55–42.23) | 38.75 (32.85–42.70) | 0.705 | 37.95 (34.20–40.85) | 36.60 (31.93–39.35) | 38.90 (34.80–41.50) | 0.041[*] |
| 24 h urinary protein (g/d) | 1.15 (0.52–2.89) | 1.41 (0.78–3.26) | 0.89 (0.30–2.29) | 0.105 | 1.11 (0.37–1.78) | 1.57 (0.85–2.70) | 0.77 (0.31–1.50) | 0.010[*] |
| TG (mmol/L) | 1.46 (0.93–2.24) | 1.45 (0.94–2.52) | 1.46 (0.91–2.12) | 0.693 | 1.22 (0.74–1.94) | 1.54 (1.13–2.50) | 1.03 (0.63–1.57) | 0.001[*] |
| TC (mmol/L) | 4.92 (4.06–5.77) | 5.02 (4.19–5.77) | 4.84 (3.97–5.91) | 0.797 | 4.81 (4.27–5.82) | 5.06 (4.43–6.04) | 4.70 (4.24–5.53) | 0.176 |
| eGFR (ml/min 1.73 m² ) | 82.75 (57.20–105.90) | 60.20 (45.45–91.33) | 88.55 (72.40–112.03) | 0.005[*] | 105.30 (70.10–118.10) | 66.27 (37.28–87.53) | 115.40 (100.00–121.70) | <0.001[*] |
| Oxford classification (MEST-C) | | | | | | | | |
| M1 | 54 (71.1) | 25 (78.1) | 29 (65.9) | 0.246 | 56 (73.7) | 16 (66.7) | 40 (76.9) | 0.345 |
| E1 | 32 (42.1) | 18 (56.3) | 14 (31.8) | 0.033[*] | 35 (46.1) | 11 (45.8) | 24 (46.2) | 0.979 |
| S1 | 55 (72.4) | 26 (81.3) | 29 (65.9) | 0.140 | 55 (72.4) | 17 (70.8) | 38 (73.1) | 0.839 |
| T1–T2 | 20 (26.3) | 12 (37.5) | 8 (18.2) | 0.059 | 16 (21.1) | 10 (41.7) | 6 (11.5) | 0.003[*] |
| C1–C2 | 24 (31.6) | 7 (21.9) | 17 (38.6) | 0.121 | 30 (39.5) | 8 (33.3) | 22 (42.3) | 0.457 |

**Notes.**

Data are expressed as numbers and percentages in non-continuous variables, as means ±standard deviation in parametric continuous variables, and as median and interquartile range in nonparametric continuous variables.

BUN, blood urea nitrogen; SUA, serum uric acid; TG, triglycerides; TC, total cholesterol; eGFR, estimated glomerular filtration rate; M1, mesangial hypercellularity; E1, endocapillary hypercellularity; S1, segmental glomerulosclerosis; T1–T2, tubular atrophy/interstitial fibrosis >25%; C1–C2, presence of crescent.

*Two-tailed $P < 0.05$.

TA-SUA with baseline age and TG (Figs. 2A and 2D) were not significant in male patients (all $P > 0.05$).

Among the analyzed serum complement features, the TA-SUA level showed positive correlations with serum C3 and serum C4 (Table 3), but there were not significant in male and female patients with IgAN (Fig. S3).
**Table 3 Correlations between TA-SUA levels and clinicopathological characteristics.**

| | TA-SUA (µmol/L) overall | | TA-SUA (µmol/L) | | | |
| | Correlation coefficient (r) | P | Male | | Female | |
| | | | r | P | r | P |
|---|---|---|---|---|---|---|
| Baseline age | 0.170 | 0.036* | 0.064 | 0.581 | 0.267 | 0.020* |
| BUN | 0.488 | <0.001* | 0.242 | 0.035* | 0.557 | <0.001* |
| TG | 0.359 | <0.001* | 0.077 | 0.508 | 0.461 | <0.001* |
| eGFR | −0.627 | <0.001* | −0.414 | <0.001* | −0.717 | <0.001* |
| Serum C3 | 0.187 | 0.021* | 0.113 | 0.329 | 0.142 | 0.222 |
| Serum C4 | 0.170 | 0.037* | 0.209 | 0.070 | 0.112 | 0.341 |
| 24 h urinary protein | 0.345 | <0.001* | 0.273 | 0.017* | 0.315 | 0.006* |

Notes.
Spearman's correlation analysis.
TA-SUA, time-averaged serum uric acid; BUN, blood urea nitrogen; TG, triglycerides; eGFR, estimated glomerular filtration rate.
*Two-tailed $P < 0.05$.

## Association of TA-SUA level with long-term renal outcomes in IgAN patients

A total of 19 patients with IgAN suffered from renal endpoint events during the follow-up (58.08 ± 23.51 months). As shown in Fig. 3A, the Kaplan–Meier renal survival curve indicated that the accumulative survival rate for the primary outcome was significantly lower in patients with TA-SUA levels in the highest quartile than in those with TA-SUA levels in the lower quartiles ($P < 0.001$). Moreover, from the Kaplan–Meier renal survival analysis comparing the male and female groups with persistent hyperuricemia and normo-uricemia, the renal survival rate in male patients with persistent hyperuricemia was significantly lower than that in male patients with normo-uricemia (Fig. 3B), and this difference was also observed in female patients (Fig. 3C).

## Predictive value of the TA-SUA level for renal prognosis

We applied a Cox proportional hazards model to determine risk factors for the composite endpoint. The proportional hazards assumption based on Schoenfeld residuals was shown in Fig. S4. The results showed that the risk proportional assumption in Cox proportional hazard model was congruent ($P > 0.05$). Univariate Cox regression analysis identified the TA-SUA level (hazard ratio (HR) = 1.014, 95% confidence interval [CI]: 1.009–1.020, $P < 0.001$), hypertension (HR = 3.358, $P = 0.002$), hemoglobin level (HR = 0.960, $P = 0.001$), hyperuricemia (HR = 3.543, $P = 0.003$), baseline eGFR (HR = 0.961, $P < 0.001$), 24-h urinary protein level (HR = 1.283, $P = 0.004$), and T1–T2 (HR = 12.676, $P < 0.001$) as significantly associated with a poorer prognosis in patients with IgAN. On multivariable regression analysis including all clinical characteristics significantly associated with renal survival, the TA-SUA level (HR = 1.011, 95% CI [1.005–1.017], $P = 0.001$) and T1–T2 (HR = 4.893, $P = 0.001$) remained independent risk factors for renal dysfunction in patients with IgAN, but hemoglobin level (HR = 0.973, $P = 0.007$) was a protective factor. (Table 4). Furthermore, multivariable regression analysis was used to evaluate independent prognostic factors for the progression of IgAN to renal failure in men and women separately. The results confirmed that the TA-SUA level was a significant

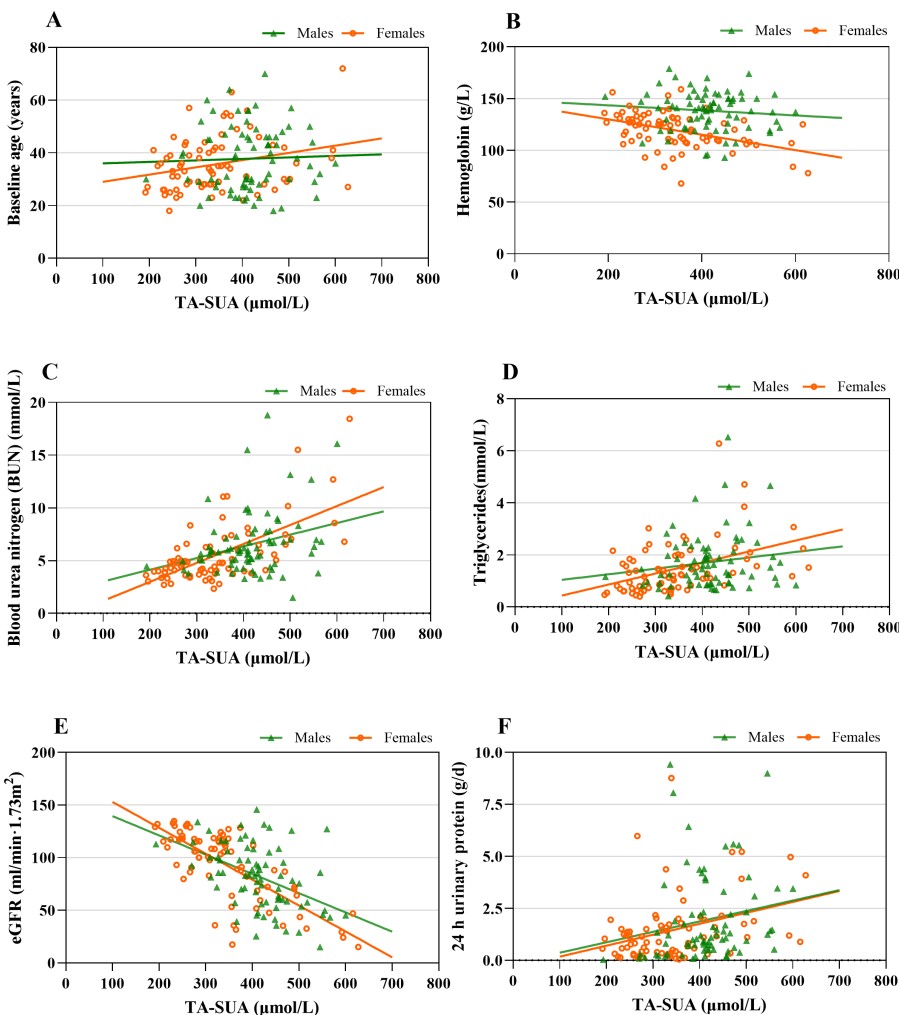

**Figure 2   The associations between time-averaged serum uric acid (TA-SUA) levels and clinical data at the time of renal biopsy among male and female patients with IgA nephropathy.** TA-SUA correlated with (A) baseline age (male: $r = 0.064$, $P = 0.581$; female: $r = 0.267$, $P = 0.020^*$); (B) hemoglobin (male: $r = -0.090$, $P = 0.439$; female: $r = -0.470$, $P < 0.001^*$); (C) blood urea nitrogen (male: $r = 0.242$, $P = 0.035^*$; female: $r = 0.557$, $P < 0.001^*$); (D) triglycerides (male: $r = 0.077$, $P = 0.508$; female: $r = 0.461$, $P < 0.001^*$); (E) estimated glomerular filtration rate (eGFR) (male: $r = -0.414$, $P < 0.001^*$; female: $r = -0.717$, $P < 0.001^*$); and (F) 24-h urinary protein (male: $r = 0.247$, $P = 0.017^*$; female: $r = 0.315$, $P = 0.006^*$). Spearman's correlation analysis; $^*$Two-tailed $P < 0.05$.

independent predictor of renal outcomes in both male (HR = 1.013, 95% CI [1.002–1.024], $P = 0.025$) and female (HR = 1.010, 95% CI [1.001–1.020], $P = 0.033$) patients with IgAN (Table 5).

Next, we examined the predictive value of the TA-SUA level for the progression of IgAN by ROC curve analysis. Because the standard basic uric acid level differs for male and female patients, we performed ROC curve analyses for male and female patients separately. As shown in Fig. 4, the AUC values reflecting the predictive value of TA-SUA in male and female patients were 0.934 and 0.768, respectively. The corresponding optimal cutoff value

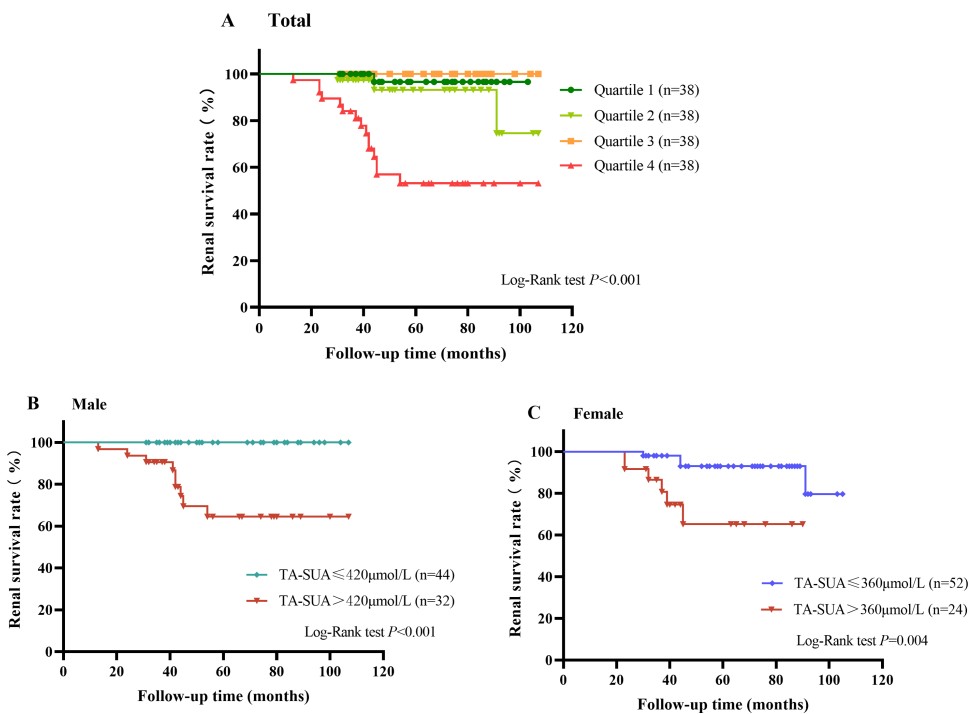

**Figure 3** **Kaplan–Meier curve for the renal survival rate in patients with IgA nephropathy according to the time-averaged serum uric acid (TA-SUA).** (A) of quartiles; Quartile 1: TA-SUA < 311.5 μmol/L; Quartile 2: 311.50 μmol/L < TA-SUA < 376.02 μmol/L; Quartile 3: 376.03 μmol/L < TA-SUA < 444.46 μmol/L; Quartile 4: TA-SUA > 444.46 μmol/L. (B) in males; and (C) in females. The event-free survival for the doubling of the baseline serum creatinine level or end-stage renal disease.

for the TA-SUA level for predicting IgAN progression was 451.38 μmol/L in men with a high sensitivity (100.0%) and specificity (77.6%) (Fig. 4A), and this value was 492.83 μmol/L in women with a low sensitivity (60.0%) but high specificity (98.5%; Fig. 4B), as calculated by obtaining the best Youden index. The Kaplan–Meier analysis indicated that the renal survival rate for the primary outcome was significantly lower in patients with TA-SUA > optimal cutoff value (451.38 μmol/L for men and 492.83 μmol/Lfor women) than in those with TA-SUA ≤optimal cutoff (both $P < 0.001$) (Figs. 4C and 4D).

## DISCUSSION

The concentration of SUA, the end product of purine metabolism by xanthine oxidoreductase, is correlated with many recognized cardiovascular risk factors. Some studies have emphasized that SUA at high levels forms urate crystals that deposit in the renal tubules and interstitium, leading to kidney fibrosis and renal failure (*Su et al., 2020*; *Viggiano et al., 2018*). Hyperuricemia has been shown to be correlated with IgAN and identified as an independent predictor of an unfavorable renal outcome in Chinese adult patients with IgAN (*Le et al., 2012*). However, whether an elevated SUA level or its persistence can provide an independent prognostic factor in IgAN remained unknown. In this retrospective study of 152 IgAN patients, we comprehensively evaluated the correlation

**Table 4 Risk factors for renal endpoint determined by univariate/multivariate Cox hazard analysis in patients with IgAN.**

| Factors | Univariate | | Multivariate | |
|---|---|---|---|---|
| | HR (95% CI) | P value | HR (95% CI) | P value |
| Baseline age | 0.982 (0.938–1.026) | 0.436 | | |
| Hypertension | 3.358 (1.569–7.881) | 0.002* | 1.561 (0.380–4.219) | 0.380 |
| Hemoglobin | 0.960 (0.937–0.983) | 0.001* | 0.973 (0.953–0.993) | 0.007* |
| Hyperuricemia | 3.543 (1.542–8.141) | 0.003* | 3.109 (0.824–11.726) | 0.094 |
| Baseline eGFR | 0.961 (0.946–0.977) | <0.001* | 0.993 (0.971–1.014) | 0.493 |
| 24 h urinary protein | 1.283 (1.081–1.522) | 0.004* | 1.097 (0.851–1.414) | 0.475 |
| M1 | 0.667 (0.252–1.764) | 0.414 | | |
| E1 | 1.342 (0.545–3.305) | 0.522 | | |
| S1 | 3.674 (0.847–15.929) | 0.082 | | |
| T1–T2 | 12.676 (4.690–34.260) | <0.001* | 4.893 (1.861–12.869) | 0.001* |
| C1–C2 | 1.083 (0.426–2.752) | 0.867 | | |
| TA-SUA (μmol/L) | 1.014 (1.009–1.020) | <0.001* | 1.011 (1.005–1.017) | 0.001* |

**Notes.**

CI, confidence interval; HR, hazard ratio; eGFR, estimated glomerular filtration rate; M1, mesangial hypercellularity; E1, endocapillary hypercellularity; S1, segmental glomerulosclerosis; T1–T2, tubular atrophy/interstitial fibrosis >25%; C1–C2, presence of crescent; TA, time-averaged.

*Two-tailed $P < 0.05$.

**Table 5 Multivariate Cox hazard analysis in male and female patients with IgAN.**

| Factors | Male | | Female | |
|---|---|---|---|---|
| | HR (95% CI) | P value | HR (95% CI) | P value |
| BUN | 0.924 (0.763–1.119) | 0.417 | – | – |
| Serum albumin | – | – | 0.913 (0.771–1.081) | 0.290 |
| Baseline eGFR | 0.974 (0.935–1.014) | 0.198 | 0.997 (0.969–1.026) | 0.850 |
| 24 h urinary protein | 1.456 (1.002–2.116) | 0.049* | – | – |
| T1–T2 | 2.593 (0.971–6.920) | 0.057 | 5.270 (1.188–23.377) | 0.029* |
| TA-SUA | 1.013 (1.002–1.024) | 0.025* | 1.010 (1.001–1.020) | 0.033* |

**Notes.**

CI, confidence interval; HR, hazard ratio; BUN, blood urea nitrogen; eGFR, estimated glomerular filtration rate; T1–T2, tubular atrophy/interstitial fibrosis >25%; TA-SUA, time-averaged serum uric acid.

*Two-tailed $P < 0.05$.

between TA-SUA levels and clinical and histopathological features. Our analyses showed that the levels of TA-SUA were independent risk factor associated with IgAN progression. Further, persistent hyperuricemia influenced outcomes in patients of both sexes, but its impact is greater in females than in males. The sex-specific prevalence of IgAN has been shown to vary geographically, with a reported male:female ratio of almost 1:1 in Asian populations, compared to 6:1 in Europe and the United States (*Goto et al., 2009*; *Duan et al., 2013*; *Geddes et al., 2003*). This ratio in the present study was consistent with the previously reported ratio for Asian populations, which may be attributed to the patients' geographical proximity and comparable diets.

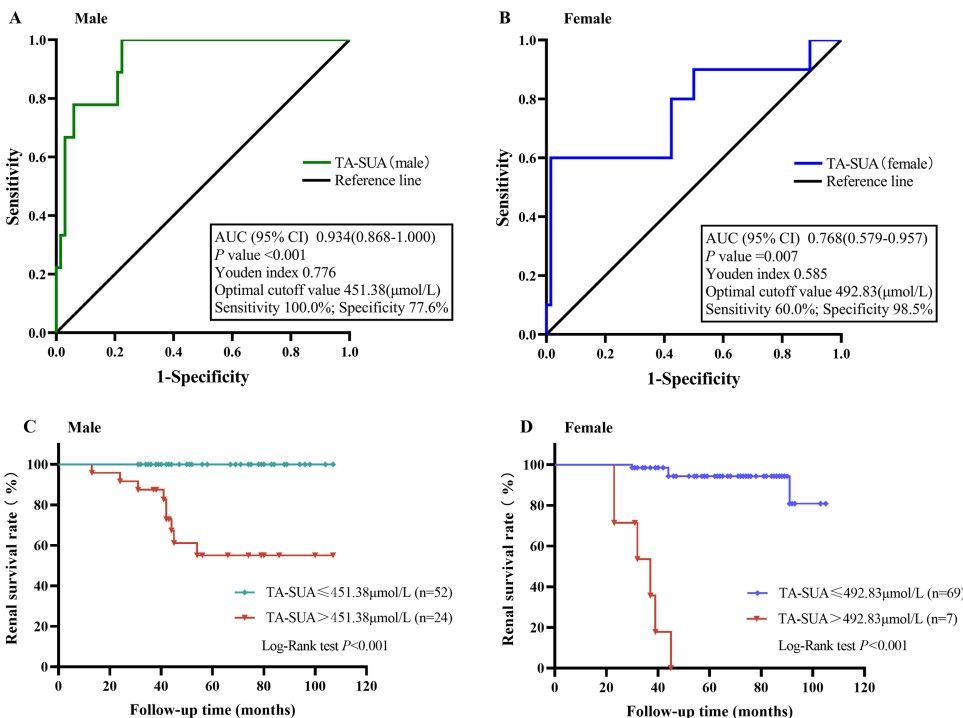

**Figure 4** **The value of time-averaged serum uric acid (TA-SUA) levels in predicting renal prognosis and Kaplan-Meier analysis.** ROC curve analysis evaluated the predictive value of the TA-SUA level for poor renal outcomes. (A) in males and (B) in females with IgA nephropathy. Kaplan-Meier curve for the renal survival rate between two groups (TA-SUA > optimal cutoff value *Vs.* TA-SUA ≤ optimal cutoff). (C) in males and (D) in females with IgA nephropathy.

Data from several studies have suggested that patients with IgAN are prone to hyperuricemia, which is closely correlated with blood pressure, TG level, 24-hour urinary protein, and renal function (*Lu et al., 2021*; *Yi et al., 2021*), and this was generally observed in the present study. We measured the average level of SUA during the follow-up period and confirmed that patients with higher TA-SUA levels were more likely to have hypertension and higher levels of baseline SUA, TG, BUN and proteinuria, as well as worse renal function. The SUA level is closely linked to accumulation of visceral fat (omentum majus, perirenal, *etc.*) and increases further with the increase in serum TC and TG (*Hikita et al., 2007*). Positive associations previously were found between the TG-glucose(TyG) index and SUA and between the TyG index and hyperuricemia in adults with hypertension (*Yu et al., 2022*). Our study also confirmed that the TA-SUA level showed a strong positive correlation with TG ($r = 0.359$, $P < 0.001$), especially in female patients with IgAN ($r = 0.461$, $P < 0.001$). For almost two decades, experimental models have been used to examine the nephrotoxic effects of uric acid (UA). A comprehensive review of these studies described that the toxic effects of UA include the induction of endothelial dysfunction with impairment of nitric oxide production, activation of the RAAS, and increased production of pro-inflammatory cytokines (*Weaver Jr, 2019*), leading to hypertension and eventually decreased renal function.

In the present study, the TA-SUA level also correlated with the levels of serum C3 and C4. A previous study suggested that hyperuricemia might induce the expression of complement components by activating the proinflammatory nuclear factor-kappa B (NF-$\kappa$B) signaling cascade (*Spiga et al., 2017*). IgAN is an autoimmune disease associated with complement activation, and extensive research has shown that hyperuricemia and the deposition of C3 are independent risk factors for IgAN progression (*Caliskan et al., 2016*; *Nam et al., 2020*). However, the effects of serum C3 and C4 have been a controversial and much disputed subject within the field of IgAN. It was previously observed that an increase in serum C4, as well as a decrease in C3, was an important outcome determinant for patients with IgAN (*Pan et al., 2017*). Another study demonstrated that a decreased C3/C4 ratio corresponds to poor renal outcomes and the potential to benefit from aggressive immunosuppressive therapies (*Zhang et al., 2020*). However, the effect of the complement component on the prognosis of IgAN was not further explored in our study. Only a positive correlation was found between the TA-SUA level and serum C3 and C4 levels, but the correlation was not significant in either the male or female population separately. Therefore, a larger sample size for complement C3 and C4 measurement is needed to gain evidence for guiding subsequent immunotherapy.

A recent meta-analysis including 25 studies with a total of 6048 IgAN patients suggested that hyperuricemia worsens the renal prognosis by aggravating the clinical outcomes and pathological condition of IgAN (*Geng et al., 2022*). Our results also indicated that a baseline hyperuricemia was significantly associated with a higher risk of developing D-SCr or ESRD in univariate Cox hazard analysis, but it was not a significant independent predictor of renal outcomes in patients with IgAN through multivariate Cox hazard analysis. In further subgroup analysis according to gender, previous studies suggested that the baseline SUA level is a predictor of IgAN in females but not in males (*Nagasawa et al., 2016*; *Oh et al., 2020*). Considering the uncertainty of single measurement of SUA, we further evaluated the TA-SUA level during follow-up and observed that the TA-SUA level was a significant independent predictor of renal outcomes in patients with IgAN, with optimal cutoff values of 451.38 $\mu$mol/L (area under the curve (AUC) = 0.934) for men and 492.83 $\mu$mol/L (AUC = 0.768) for women, even after adjustment for confounding factors. Therefore, our study indicated that the TA-SUA offers a better predictor of renal prognosis than the baseline SUA level. However, SUA is reported to be related to more severe renal histopathology and worse prognosis in female IgAN patients (*Choi et al., 2021*; *Oh et al., 2020*). At present, it is not clear why the serum uric acid level of women is more important than that of men, but it has been reported that estrogen can inhibit urate transporter 1, reduce serum uric acid level and promote urinary acid excretion (*Takiue et al., 2011*). Estrogen plays a vital role in renal protection. Estrogen negatively regulates TGF-synthesis; estrogen deficiency and ovariectomy will accelerate the progression of glomerular injury (*Kelimu et al., 2017*). Endogenous estradiol can reduce serum uric acid level (*Mumford et al., 2013*). This may lead to the observed gender difference in IgAN. The average age of women in this study is 34 years old, and the protective effect of estrogen on renal function may be the reason why the optimal critical value (492.83$\mu$mol/L) of women is higher than that of men.

Tubular atrophy and interstitial fibrosis have been demonstrated to be essential independent risk factors for the progression of IgAN (*Trimarchi et al., 2017*; *Woo et al., 2016*). Consistently, in the present study, T1–T2 was a risk factor for poor prognosis of IgAN, even with relevant adjustment in the multivariable analysis models, and the frequency of T1–T2 among patients with higher levels of TA-SUA was also higher, especially in women. Recent research has established that hyperuricemia and hypertriglyceridemia, which are prevalent in patients with CKD, are independent risk factors for moderate tubular atrophy/interstitial fibrosis (*Liu et al., 2021*), and by multivariate logistic regression analysis, only tubular atrophy/interstitial fibrosis (T1–T2) (HR = 3.969, $P = 0.008$) was significantly associated with hyperuricemia in IgAN (*Ruan et al., 2018*). This suggests that we can combine the pathological characteristics and clinical follow-up data into a model for predicting the progression of IgAN.

Controversy persists regarding whether treatment of hyperuricemia will benefit the prognosis of IgAN. The recent large trial by Kimura et al. suggests that compared to placebo, febuxostat did not mitigate the decline in kidney function among patients with stage 3 CKD and asymptomatic hyperuricemia. One positive response was a significant improvement in the eGFR slope in patients who had SCr levels less than the median for the group, but no change was observed in other groups (*Kimura et al., 2018*). More data and research are needed to further explore the benefits of such treatments.

The present study has several limitations. First, our analyses could not resolve the potential effects of all hidden biases and confounding factors. For example, SUA levels are affected by particular foods, but data regarding patients' intake of these foods were not available. Second, our study lacked data regarding patients' use of UA-lowering medications; the effects of these agents on the kidneys have emerged as a topic of considerable interest recently. The therapeutic relevance of these studies, however, remains a controversial subject. The main point of contention rests in the fact that hyperuricemia is to be expected in patients with reduced kidney function, and thus, it is difficult to prove that increased levels of UA are a cause of decreased kidney function. Third, this was a single-center study with a relatively small sample size. Therefore, a prospective multi-center study is required to further validate our findings. Fourth, this study only included the clinical data of patients diagnosed as IgAN by renal biopsy, and did not set up a cohort of non-IgAN patients as a control, which is also the deficiency of this study. Finally, because urine protein was not taken as a routine review item in the later period of follow-up, the urine protein data of 152 patients with IgAN were seriously missing, so this study only collected relatively complete urine protein data within 6 months of onset. we will continue to collect relevant data as much as possible and pay attention to the changes of proteinuria.

## CONCLUSIONS

The results of the present study show that the level of TA-SUA is associated with the levels of serum TG, C3, and C4 as well as renal function and the pathological severity of IgAN. According to our data, hyperuricemia significantly promotes IgAN progression, especially

in female patients, whereas its persistence in significant amounts is an independent risk factor for renal function loss both in male and female patients. We recommend that these findings be considered in treatment planning and in the design of prospective therapeutic trials.

### Funding
This work was supported by the Fujian Provincial Science and Technology Plan Project (No. 2021Y2005). The funders had no role in study design, data collection and analysis, decision to publish, or preparation of the manuscript.

### Grant Disclosures
The following grant information was disclosed by the authors:
Fujian Provincial Science and Technology Plan Project: 2021Y2005.

### Competing Interests
The authors declare there are no competing interests.

### Author Contributions
- Mengjie Weng conceived and designed the experiments, performed the experiments, prepared figures and/or tables, authored or reviewed drafts of the article, and approved the final draft.
- Binbin Fu conceived and designed the experiments, performed the experiments, prepared figures and/or tables, authored or reviewed drafts of the article, and approved the final draft.
- Yongjie Zhuo performed the experiments, prepared figures and/or tables, and approved the final draft.
- Jiaqun Lin performed the experiments, prepared figures and/or tables, and approved the final draft.
- Zhenhuan Zou performed the experiments, prepared figures and/or tables, and approved the final draft.
- Yi Chen analyzed the data, authored or reviewed drafts of the article, and approved the final draft.
- Jiong Cui performed the experiments, prepared figures and/or tables, and approved the final draft.
- Guifen Li analyzed the data, authored or reviewed drafts of the article, and approved the final draft.
- Caiming Chen analyzed the data, authored or reviewed drafts of the article, and approved the final draft.
- Yanfang Xu analyzed the data, authored or reviewed drafts of the article, and approved the final draft.
- Dewen Jiang analyzed the data, authored or reviewed drafts of the article, and approved the final draft.

- Jianxin Wan conceived and designed the experiments, authored or reviewed drafts of the article, and approved the final draft.

## Human Ethics

The following information was supplied relating to ethical approvals (*i.e.*, approving body and any reference numbers):

Informed consent approved by the local ethical committee has been obtained from each patient after a full explanation of the purpose and nature of all procedures used. The study adhered to all relevant tenets of the Declaration of Helsinki and was approved by the Ethics Committee of the First Affiliated Hospital of Fujian Medical University (MTCA, ECFAH of FMU [2015] 084-2).

## Data Availability

The raw data are available in the Supplemental Files.

## Supplemental Information

Supplemental information for this article can be found online at http://dx.doi.org/10.7717/peerj.17266#supplemental-information.

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
