# Peer review of "Association of time-averaged serum uric acid level with clinicopathological information and long-term outcomes in patients with IgA nephropathy"

_PeerJ, doi:10.7717/peerj.17266_

## Round 0.1 · original submission · Major Revisions

Please check the statistical analysis in accordance with reviewer 2 comments and add information to the DIscussion and Introduction sections, as suggested.

Reviewer 1 ·

Basic reporting

The work is novel and has clinical significance. The abstract is well presented. Methods need to be improved further. The results are well presented

Experimental design

1. The authors should have included only patients with IgAN. The authors have not included any patients without IgAN. The best way to assess the risk factors is to create a contingency table or cross table with the patients with and without IgAN in the row-wise category and the patients with persistent hyperuricemia and normal uricemia in the column-wise category. This was not done in the study. With this cross-table, authors will be able to analyze the data with the Chi-square test and also apply the odds ratio. Now this is not possible as the authors do not have a patient population without IgAN. Probably, authors need to state this as a limitation of the study.
2. The authors need to incorporate the STROBE flow chart, including how many patients were excluded and reasons for exclusion.

Validity of the findings

The findings are valid with the current data analysis.

Additional comments

None

·

Basic reporting

1. In the baseline characteristics (Table 1), how is the blood glucose level / diabetic history of the patients? They are related to metabolism together with hyperuricemia. How is the medication history among the study population? Some medications might affect time-averaged SUA. How is the CKD stage?

2. How is the rate of change in proteinuria in this cohort? Please add more details in the follow-up description, i.e.: TA-proteinuria, relapse status.

Experimental design

1. The authors performed ROC curve analysis for TA-SUA in predicting renal prognosis (Figure 3). However, it is not considered in the current analysis that the renal disease status also changes over time. Have the authors tried time-dependent ROC? Or further Kaplan-Meier analysis could be performed in this cohort between two groups (TA-SUA>= optimal cutoff value Vs. TA-SUA< optimal cutoff).

2. Since TA-SUA is a time dependent covariate, the proportional hazards assumption based on Schoenfeld residuals is suggested to be checked for verifying the result of the cox regression analysis.

Validity of the findings

1. In the Kaplan-Meier curve (Figure 2), please add the number of each group at risk.

2.The authors found that TA-SUA level also was an independent predictor of renal outcome in patients with IgAN, with optimal cutoff values of 451.38 μmol/L (area under the curve [AUC]=0.934) for men and 492.83 μmol/L (AUC=0.768) for women. However, SUA is reported to be related to more severe renal histopathology and worse prognosis in female IgAN patients (PMID 33925441, 31936416). Why is the cutoff value of TA-SUA higher in female than male? Please add the possible explanations in the discussion.

3. Serum immunoglobulins and pathological immune filtration features are included in the raw data but not described in the manuscript, please add the related results in the baseline characteristics.

Additional comments

1. In the abstract, how is the prognosis defined? How is the study designed, retrospective or prospective? Please be specific.

2. Line 166, Supplemental Figure 3 should be Supplemental Figure 2.

3. Table 3, please add the correlations between TA-SUA and clinicopathological characteristics in the female and male subgroup.

Reviewer 3 ·

Basic reporting

This is a well written paper with only occasional typos and excellent references. The structure of the paper is good with sufficient raw data.

Experimental design

Well designed study. I have a few simple questions.
1.Why did the study only involve patients biopsied after 2012?
2. Why was the follow-up period concluded in 2021?
Both of these questions are posed because of the small number of patients in the study.
3. The authors indicate that questions regarding medications were asked at each of the 6 month clinic visits but information regarding the use of uric acid meds was not available. Please explain.
4. The authors report on the presence of hematuria in a number of places with the first mention (line 78) being focused on a history of macroscopic hematuria. Please clarify whether this was the case elsewhere in the paper, including the tables.
5. The authors state that normouricemia was defined when both baseline and TA SUA levels were normal. How did they define patients when one of the measurements was normal and the other was high?
6. Did the authors require a minimum number of SUA levels in individual patient for the determination of a TA-SUA?

Validity of the findings

No Comment.

Additional comments

The authors have appropriately used the standard definitions for HU in men and women, based on the normal range seen in population studies but it is interesting that the optimal cut-off point for the TA-SUA level for predicting CKD progression was higher in women (492) than in men (451).....lines 198-199. This begs the question whether the normal levels quoted for the two sexes has any direct relationship to SUA toxicity levels.The authors might wish to discuss this????

Reviewer 4 ·

Basic reporting

The manuscript under review presents a robust and well-crafted document with an innovative perspective on the predictive nature of time-averaged serum uric acid level on outcome in patients with IgA nephropathy. The writing is succinct and clear, with minor grammar revision, featuring a well-supported literature introduction and solid data.
1. Grammer- well written text, minor revisions were suggested in the annotated manuscript.
2. Introduction will benefit from expanding on the current treatments that are available and how their mechanism effects renal measurements.
3. Line 45, state what is your reasoning to note that it cannot reflect the longitudinal variation it?
4. Line 48-56, would be nice to state what is your hypothesis, what did you expect to find?

Experimental design

nicely outlined primary research within Aims and Scope of the journal, the Research question is well defined (line48-56).
Methods were described well and were relevant to the data presented, please see comments in the manuscript text.
5. Line 72, RAAS inhibitor, explain the effect/ mechanism of action in the introduction.
6. Line 85, Definitions- suggesting using a different title; parameter characterization?
7. Line 91 would have been nice to see an actual example of calculation in the supplementary figures.
Results data presented answered the authors specific hypothesis and had significant predictive benefit which is both relevant & meaningful.
8. Line 166, you mentioned Supplementary figure 3, did you mean supplementary figure 2?

Validity of the findings

Conclusions are well stated, linked to original research question and well supported with literature.

Annotated reviews are not available for download in order to protect the identity of reviewers who chose to remain anonymous.

---

## Round 0.2 · Minor Revisions

Please clarify the points related to urate‐lowering medications, history of hyperuricemia and monitoring of proteinuria.

Reviewer 1 ·

Basic reporting

No comment

Experimental design

No comment

Validity of the findings

No comment

Additional comments

The research work contributed by the researchers is novel. The authors have thoroughly revised, addressing each reviewer's comments. The author's research work is appreciable.

·

Basic reporting

Most questions raised were answered to the point. The writing is clear and concise. The work could benefit more if the following details could be added.

1. Had the urate‐lowering medications been used in the patients? Did the patients have a history of hyperuricemia before biopsy?

2. The authors replied that: “urine protein was not taken as a routine review item in the later period of follow-up”, however, the follow-up plan in the method part was still described as “All patients were followed up through regular visits at intervals of 6 months. Serum uric acid, proteinuria, …… were tested at every visit.” Please modify it.

Experimental design

All questions raised were answered to the point. The reviewer appreciate the authors's hard work.

Validity of the findings

Conclusions are well stated, linked to original research question and well supported with literature.All questions raised were answered to the point. The reviewer appreciate the authors's hard work.

Additional comments

NoneAll questions raised were answered to the point. The reviewer appreciate the authors's hard work.

Reviewer 3 ·

Basic reporting

No comment

Experimental design

No comment

Validity of the findings

No comment

Additional comments

No comment

Reviewer 4 ·

Basic reporting

The authors' adept incorporation of the suggested corrections has greatly enhanced the clarity and comprehension of both the manuscripts and the associated data. These revisions not only elevate the overall quality of the material but also provide deeper insights into the correlation between time-averaged serum uric acid levels and clinicopathological characteristics, as well as long-term outcomes in patients with IgA nephropathy. Such insights hold significant value for researchers and clinicians, enriching our understanding and potentially guiding future diagnostic and therapeutic strategies in this patient population.

Experimental design

review comments were integrated nicely

Validity of the findings

review comments were integrated nicely

Additional comments

review comments were integrated nicely

---

## Round 0.3 · accepted · Accept

The authors responded satisfactorily to the previous comments.